# Cultivated Manatee Meat Aiding Amazon Biodiversity Conservation: Discussing a Proposed Model

Ana Flavia S. Abrahao [1,*] , Joao Paulo F. Rufino [2], Germano Glufke Reis [1] and Alexandre Cabral [3]

1   Management School, UFPR Federal University of Parana, Av. LothárioMeissner, 632, Jd. Botanico, Curitiba 80210170, Brazil
2   Department of Animal and Plant Production, Agrarian Sciences Faculty, Federal University of Amazonas, Manaus 69080900, Brazil
3   Policy Department, The Good Food Institute Brazil, Rio de Janeiro 22031020, Brazil
*   Correspondence: anaflavia.abrahao@gmail.com; Tel.: +353-083-378-2346

**Abstract:** Cultivated meat (CM) is a disruptive technology that provides an alternative to animal protein. In this context, the Amazon manatee (*Trichechus inunguis*) emerges as an important case. Although it is illegal to hunt this large mammal, its meat continues to be consumed, causing several threats to its natural habitat. The aim of this study is to explore the impacts of introducing the Amazon manatee CM into the traditional meat value chain as a tool to aid the biodiversity of the Amazon Basin. Thus, we developed a strengths, weaknesses, opportunities, and threats matrix from the content analysis of 11 interviews conducted between October 2021 and May 2022. The interviewees were experts in different fields, ranging from financial analysts of novel food technologies to biologists, researchers, and others. We presented the theme of illegal hunting and its consequences during the interviews, followed by the CM process, and explained how the royalties from the sale of this innovative product could help to preserve Amazon biodiversity through the proposal of a new business model. The main findings suggest that the proposed model would produce good results, but the threat of a rebound effect from the consumption of wild animals was mentioned in most responses, especially by actors involved in conservation. The strengths and opportunities of this disruptive narrative mainly focused on preserving biodiversity and promoting environmental awareness, combining the conservation of wildlife and the consumption of novel food. The weaknesses included the lack of knowledge and the non-existent market. This framework is relevant for policymakers, nongovernmental organizations, and researchers seeking to improve the sustainability not only of the species found in the Amazon, but also around the world.

**Keywords:** cultivated meat; wildlife trade; Amazon; biodiversity; business model





## 1. Introduction

Biotechnology processes based on tissue engineering and stem cell multiplication have made it possible to address a disruptive issue that might be found in all equations for the mitigation of climate change. These processes involve moving away from obtaining protein for human consumption through livestock and opting for cultivated meat (CM). The production of meat from a cell culture establishes a disruptive approach in many layers of this process. One of the most important of these layers is food safety and involves the donor animal's tissue [1,2], shifting the focus from big ranchers, with large herds, to a small group of cell donor individuals. The use of fewer scarce natural resources, such as land and water [3,4], qualifies this case as an object of study.

Meat production based on traditional species used in livestock (poultry, cattle, pigs, sheep, and fish) can be reviewed from the CM supply of the same species, thereby reducing the pressure on these supply chains to meet the growing demand for food [3,4]. This enables a reduction in the deforestation caused by the need for new pastures and grains required to feed all these animals. However, the meat of wild species, when multiplied

without the animal having to be slaughtered from legal or illegal breeding, can gain other theoretical nuances. For example, it can create a virtuous circle where part of the proceeds from the sale of products is reverted to the process of preserving the species itself. El Bizri et al., [5] emphasized the need to adopt sustainable and technology-intensive models for the exploitation of resources in the Amazon, with a view to maintaining the biodiversity of the region, aiding the conservation of species and the development of the local inhabitants.

The deforestation of the Amazon due to livestock has been referred to in different studies [6–8]. Its biodiversity is in peril, not only because of this burning, but also due to poaching. The Amazon region is vast, measuring over 6,700,000 km$^2$, and throughout this area, there is a network of illegal trade in meat and other by-products of wild animals that are sold mainly in street markets, with no sanitary or safety guidelines, increasing the occurrence of zoonoses [9,10]. This kind of extractive system threatens the existence of native animals owing to reduced population density and, consequently, many other species that depend on each other face extinction. This leads to several environmental problems, such as the destruction and degradation of the natural ecosystem [5]. The CM supply of wild animals can positively address this issue since recent studies have compared the environmental impact of traditional livestock and cultivated production [3].

Conceptually, CM can be defined as meat obtained from the ex situ culture of animal cells. These cells are usually obtained through a biopsy and are then cultivated under proper conditions and with proper nutrients and energy sources in a bioreactor, resulting in complex structures similar to muscle and fat tissues. This process efficiently replaces conventional animal-based meat production and sufficient amounts can be produced to meet demand [11–14].

In the Amazon wildlife trade context described above, CM does not directly address all the challenges caused by poaching. However, through an intensive technological process, it would provide the market with an eco-friendly product, building a B-Company (a biotechnology company) from the outset. The larger its sales, the more it would contribute to the conservation of native species through royalties and investments with local institutes and pro-environmental organizations [15–17].

Previous work has shown that several environmental benefits can be achieved with the introduction of CM products. For instance, based on a life cycle assessment, Tuomisto and de Mattos [18] concluded that 99% less land will be needed to produce CM, while GHG emissions will be 78–96% lower, depending on the product (e.g., beef, poultry) and the energy used in the process. Likewise, Mattick et al. [19] and Alexander et al. [20] found that CM will demand less land use. Swartz [21] showed that CM is likely to lower global warming impacts by 17%, 52%, and 85–92%, respectively, compared with conventional chicken, pork, and beef production. However, these studies mostly take into account comparisons with industrial animal-based meat production (e.g., beef) and how CM can tackle environmental issues related to large-scale meat chains. The possible implications and benefits of CM technology for wild animal-based food systems have not yet been addressed.

However, considering what is outlined above, we advocate that there is room for disruptive innovation with regard to producing novel food from native species without the negative consequences it normally entails. On the contrary, it would help to make improvements. Therefore, the aim of the present study is to explore whether and how CM technology can be used to alleviate these conditions in the Amazon region. We propose the following research questions: (i) understand the view of specialists about the potential for applying cultivated meat technology as a conservation tool and (ii) create a business model exploring economic growth and, at the same time, the conservation of species. Our proposal is to replicate meat from wild Amazon animals. A manatee, for instance, a large aquatic mammal, the *Trichechus inunguis*, will donate its cells while still alive, and we would grow these cells in the laboratory, combine the muscle tissue with fat tissue, and put it in a tin. This final product would have a QR code on its label with all this information available

for consumers to access, beginning with the characteristics of the donor animal. This could build a bond of affective and emotional appeal because buyers could see the animal alive and swimming either in the tank where it was born or in the Amazon River. Other attractive features of this product derived from cultivated tissue are that it is sustainable, originated in the Amazon region, and has a positive environmental impact. It is expected to be sold not only in Brazil but also in foreign markets, where there is already a certain demand for delicatessens. The main idea of our proposed model is to revert part of the royalties to entities such as environmental preservation associations, NGOs that protect wild animals, and research institutes in order to develop conservancy strategies to preserve biodiversity. Another expected return is to aid the sustainable development of the region, not only in environmental but also in social terms, investing the profits in local communities to reduce poverty. These are some of the positive impacts that we believe our model could have.

It is important to mention that the manatee was chosen for this analysis because it is a well-known species and an endangered one that is illegally traded and consumed in the region. The Amazonian manatee is the largest aquatic freshwater mammal in South America, being found in the main rivers of the Amazon Basin, and is a symbol of the region. Even though hunting manatee has been illegal in Brazil since 1987, poaching has never ceased, resulting in the species becoming endangered. Consequently, several conservation institutions have striven to protect the Amazon's flora and fauna and the entire surrounding ecosystems [22–24].

The next step was to subject our theoretical production chain to the scrutiny of eleven experts in a series of semi-structured interviews and cross the information extracted from this content with what we gathered from the literature review. A SWOT analysis [25,26] was performed, allowing us to understand the strengths, weaknesses, opportunities, and threats that the interviewees saw in the model. The interviews were a means to examine the extent to which the proposed model is feasible in several respects (technical, marketing, cost, etc.) and how it could be improved for application in the Amazon context. Implications for further studies on CM and wildlife food systems also emerged.

## 1.1. Hunting Wild Animals in the Amazon

Subsistence hunting is one of the oldest human activities, and before animal domestication, it was the main way for different groups to acquire high-protein foods [27,28]. According to Pinheiro [29]. Subsistence hunting occurs when the hunter's sole purpose is to feed himself or his family. This type of hunting is still the vital dietary source of high-protein foods in many small, isolated communities in neotropical areas [27,30,31].

Culturally, the hunting of wild fauna by the inhabitants of traditional communities in the Amazon (known as caboclos) is a routine activity and an important means of subsistence and income for these people [32]. However, the underdevelopment and situations of social vulnerability that exist in most of these communities tend to encourage irrational exploitation of the fauna. In other words, subsistence hunting becomes predatory [30,33].

This overexploitation is a major cause of biodiversity loss in world wildlife and, in Brazil, affects a considerable number of native species [34], many of which are now endangered [35]. Changing the Amazon's natural environment causes a significant impact on the entire ecosystem, leading to (1) a decrease in the population density of hunted species; (2) reduced average body mass of animal populations as a result of the selection of larger animals; (3) a lower mean age at the first pregnancy of animal females; (4) an increase in the average fecundity of females; (5) fewer animals in older age groups; (6) a decrease in the future productivity of hunted populations; (7) local extinction of vulnerable species; and (8) changes in the structure of biological communities, owing to a lower representation of larger species [5,9,27].

## 1.2. Cultivated Meat

CM, also known as in vitro meat, cell-based meat, clean meat, and lab-grown meat, is meat produced by cellular agriculture using tissue engineering techniques [36,37]. It

represents a disruptive innovation with the potential to initiate a paradigm shift in the livestock industry [38]. Since CM is very close to conventional meat at the molecular level, it is expected to have the same sensory characteristics, including taste, texture, and appearance, and hence, become an almost perfect substitute [37,38].

Despite its incipient stage and the fact that it remains unknown to the general public, CM is attracting considerable attention from scientists, investors, and entrepreneurs [1,2]. The production costs are considerably high, and much technological improvement is still required, although improvements have been made and a great deal of effort has gone into making its industrialization and commercialization a reality [2,39].

CM presents some distinct advantages in relation to other alternative forms of protein. Unlike plant-based meats, which emulate meat using plant-based proteins, CM is a real animal protein [2,15]. Thus, it has a unique potential to directly replace animal products, addressing consumers' concerns regarding not only the sensory attributes of meat-like texture, juiciness, and flavor, but also traceability and safe purchase [1,40,41].

In a summarized and simplified way, to produce CM, the following stages, generally, must be followed [16,42]. First, a cell is extracted from a donor, which involves the removal and isolation of animal cells. "Cell banks" are an alternative, where after a primary animal sourcing event, they are continuously replicated in vitro, and can become a relatively stable cell source [12,42]. The second step is meat culturing. The bioprocess of meat cultivation can be divided into two phases with distinct goals: phase one, known as proliferation, which aims to obtain the maximum number of cells from the starting batch of cells; and phase two, the differentiation and maturation stage, where cells are seeded onto scaffolds, allowed to mature into the skeletal muscle cells, and influenced into maximum protein production (hypertrophy stage). Each of these phases has its own design requirements for the media, scaffolding, and bioreactors [1,43].

Studies have reported that CM is expected to become massively accessible to consumers in the medium term (10 to 15 years, with 2015 as the milestone). This estimate considers both the development of technology and consumer acceptance of final products [40,41,44,45]. However, Bryant [2] pointed out that various challenges for the wider acceptance of CM by the consumer have to be overcome. Due to the lack of knowledge and social and cultural resistance, information and education play an important role in this respect [46].

## 2. Methods and Data

### 2.1. Research Methodology

This work is an exploratory study, based on a qualitative framework used to answer the proposed research question. As qualitative research based on semi-structured interviews, there are three essential points to be followed: (1) conducting a literature review; (2) selecting interviewees with practical experience with the phenomenon under study; and (3) play an important role in this respect the phenomenon [47,48]. All these procedures were applied in this study. In particular, the interviewees were carefully selected in order to include experts with in-depth knowledge of several aspects of the theoretical structure presented above. These include experts in the fields of meat technology, Amazon conservation, traditional wildlife-based food systems in the Amazon region, conservancy institutions, the feasibility of our theoretical structure, international business/markets, manatees (conservation, reproduction, research), and pro-environment innovations (investment, market, start-ups, or economics).

### 2.2. Data Selection

Between October 2021 and May 2022, 11 professionals were interviewed in order to map the main perceptions and reflections regarding the proposed model. The main selection criterion used to choose the fields of work and, consequently, the professionals, was established by Cohen [49], considering their relevance in the Amazon context or their relevance in the CM context.

The selected interviewees presented different backgrounds, as shown in Table 1. All of them, to some extent, were experts either in CM, Amazon conservation, manatees (conservation, reproduction, research, and so on), or pro-environment innovations (investment, market, start-ups, or economics).

**Table 1.** Profiles and main characteristics of the interviewees.

| Interviewee | Current Professional Role | Criteria for Choosing the Participant | Country |
|---|---|---|---|
| I1 | Cultivated meat technology expert in the Brazilian branch of an international NGO that focuses on promoting the advancement of alternative proteins technologies | SciTech expert with experience with innovation and production of cultivated meat products. | Brazil |
| I2 | Is the managing director of the Brazilian branch of an international NGO that focuses on promoting the advancement of alternative proteins technologies | Experience with innovative alternative protein business models and new technologies. Has a broad view and understanding of the cultivated meat sector on a global basis and the opportunities and bottlenecks the sector has ahead. Has a deep knowledge of the Brazilian context for cultivated meat | Brazil |
| I3 | Is a senior biologist and researcher in a state-owned conservation institute. Is an expert in conservation technologies and policies | Experience in the conservation of endangered species in the Amazon region. Several years of experience specifically with manatees. Has knowledge about local regulations, and traditional wild-animal food systems. | Brazil |
| I4 | Is a top manager in a Brazilian venture capital firm specializing in food innovation and public policies | Has large experience with cultivated meat business models, especially in analyzing their viability and conditions for succeeding. | Brazil |
| I5 | Researcher and expert in environmental studies at a foreign university | Experience with agriculture, forest landscapes, and policies for food system sustainability. | US |
| I6 | Senior consultant and researcher in a private life cycle assessment | Experience with life cycle assessment and the best use of natural resources | Netherlands |
| I7 | Researcher and expert in business models, global value chains, and internationalization | Experience with innovative business models and business internationalization/ international management | UK |
| I8 | Researcher and expert in business models | Experience with innovative business models in the agribusiness and food systems segments | Brazil |
| I9 | Researcher and expert in business models in the food sector | Experience with innovative business models related to technology in the food sector | Brazil |
| I10 | A policymaker who has a strategic role in a bioeconomy-related government body | Experience with public policies regarding natural resources | Brazil |
| I11 | A biologist at a state-owned conservation institute and an expert in conservation technologies and policies | Experience with the conservation of endangered species | Brazil |

The semi-structured interview script consisted of six questions, subdivided into the concepts of Amazon native species, the use of CM, and the proposed business model [50]. Considering the Vergne and Wry [51] categorization guidelines, this work was divided into

categories and subcategories, presenting the quantitative relevance of the theme of each category. The categories were divided into inductive and deductive categories [52]. The deductive categories were taken from the literature and previous research [53,54], acting as a reference for the paper. The inductive categories, on the other hand, were based on the immersion of themes that have been repeated and do not lie within the theoretical scope of work evaluation.

### 2.3. The Interviews, Data Collection, and Data Analysis

At the beginning of each interview, Figure 1, which contains a design of the essence of the proposed model, was shown in all interviews by the same person to the respondents, and the main elements comprising it were explained to all of them. The figure illustrates the business proposal based on the conservation of Amazon wildlife and biodiversity using CM technology.

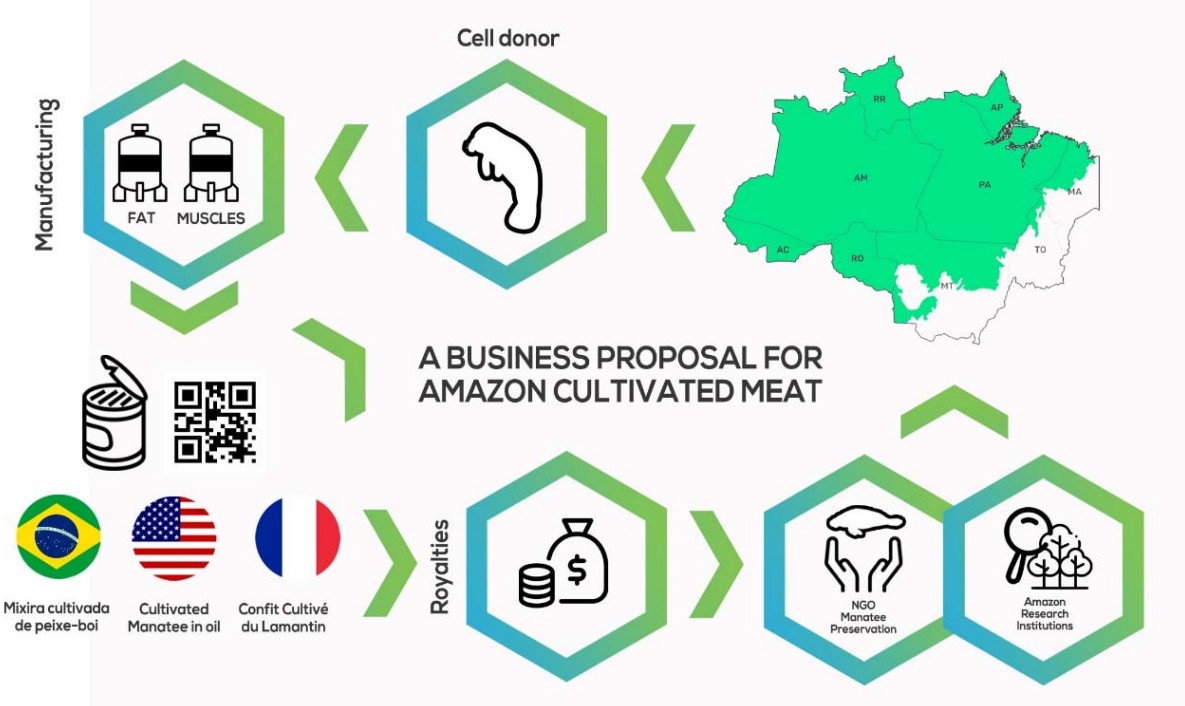

**Figure 1.** Flowchart portraying the proposed business model from the CM of Amazon manatee to the final product, its commercialization in the international market, and the royalties reverted to the conservation of the manatee species. Prepared by the authors (2021).

Using the model shown in Figure 1, we explained to the interviewees that, first, a sample of cells will be collected from a manatee housed in a controlled, comfortable, and clean environment maintained by a specialized institution. This specimen will not be captured randomly as a wild-caught animal. It is important to mention that the animal will not be slaughtered for the purpose of this collection process. A QR code with the information from the cell donor will allow full traceable monitoring, which certifies that the production chain is in compliance with local regulations regarding the use of wildlife products. The collected cell samples will be purified to achieve optimized standards for cell multiplication structures. The bioreactors will then process the samples, multiply the cells, and create the products from the command guide by the technician using state-of-the-art computers [1,43]. Finally, we emphasize that the final products are expected to give consumers an experience with a differential, as their organoleptic characteristics add to the feeling of aiding the conservation of the Amazon. In addition, we propose that since the CM product is sold in non-regional markets, i.e., outside the Amazon region, royalties would be reverted to research institutions and conservancy agencies hosted in the

Amazon to be invested in projects to protect these animals. They would mainly be research institutes and non-profit organizations with continuous and solid projects and actions for the conservation of endangered species.

All of the interviewees' responses and impressions regarding the proposed model were duly noted. Content saturation was considered when their responses, independent of one another, began to become repetitive, indicating a tendency in the line of reasoning. Therefore, when this point was reached, the interviews were ended and the analysis of the collected content began [47,48].

In addition to the content of the interviews, this work used secondary data extracted from the literature to create a database with the main evidence that ratified the statements of the interviewees and intersected to enrich the results explored at the time of analysis. The data analysis was performed using the content analysis method, with systematization and inference in accordance with the declarations and reports of the interviewees. This method is based on the researchers' ability to organize the surveyed material and map the main insights concerning the proposed theme [47,48]. The content analysis technique included the following steps: (1) pre-analysis (primary analysis of the information collected and verification of inconsistencies); (2) material exploration (highlighting the main points relevant to the study); and (3) data treatment, inference, and interpretation (crossing the information with those collected in the study). Krippendorff's steps were applied in this section for material analysis [55].

Finally, the cross-referencing of information from the insights captured during the interviews and the literature enabled the construction of a SWOT matrix [25,26] with the main opportunities, threats, strengths, and weaknesses identified in the content. This analysis provided a necessary broader view of the multifaceted and interdisciplinary research object. This object ranges from a business model in disruptive biotechnology, passing through the spillover effects of the commercialization of this technology to the conservation processes of the cell donor species. The business model is disruptive, as CM consumption has only been regulated in Singapore, and even more recently in the United States of America, as announced by the Department of Agriculture Food Safety and Inspection Service.

## 3. Results

The findings show that the perspectives of the responses varied considerably from one expertise to another. However, the responses of interviewees from similar sectors had more in common with each other, as might be expected.

The first set of questions aimed to analyze the feasibility of this project with a novel food proposal. Therefore, the interviewees were asked whether they saw more opportunities or more threats in it, as well as more weaknesses or strengths. During the interviews, they were encouraged to elaborate on their perceptions.

These results support the multifaceted and interdisciplinary characteristics of our subject, its emotional appeal, and business opportunity; for instance, the value that the positive image from the QR code carries, bringing together consumption and nature (the place where it came from). The cognitive dissonance of meat eaters would be sharply reduced in this case. Here, being able to see the welfare of an animal happily swimming in its natural habitat adds value to the product, making consumers part of the conservation process, possibly enabling them to feel special about it. This was highlighted by interviewees 6 and 5, respectively. On the other hand, at the opposite extreme, our first interviewee said that non-trivial knowledge of cell multiplication biotechnology is required, which needs to be produced on an economically viable scale, configuring a technology-based enterprise with a low number of highly specialized jobs and integrated with the conservation of a certain species.

Figure 2 summarizes what was evident in the responses of the SWOT analysis, showing the pros and cons of our proposed model from the perspective of the eleven experts.

## Opportunities

- Rain Forest and wildlife appeal;
- International exotic meat market;
- Green disruptive initiative;
- 100% sustainable;
- Stimulates altruistic attitudes;
- Amazon delicatessen;
- Great variety of meats (wild animals);
- Boosts the economy by creating and providing new jobs;
- Creates value and funds for preservation and conservation of different species;
- Seductive idea, famous actors and chefs in advertising;
- Production of logistics royalties reverted to the social and environment development of the region.

## Threats

- Rebound effect arouses curiosity about a kind of meat that is not yet available worldwide;
- Access to genetic heritage by foreign technologies;
- Patent not yet registered;
- Requires many different skills to produce;
- High-risk investment;
- Actual meat can be labelled as cultivated meat and sold on the black market.

## Strengths

- Curiosity about the "originality" of the product
- Altruistic perception on the part of consumers, creates a beneficial feeling
- Unique experience
- Trade-off: Competitive price, risks and consequences of poaching might be deterred.
- Meat from wild animals can be tastier and healthier
- Technologically, it is an easier meat to prepare than shrimp for example
- Exotic meat and "legal" consumption
- R&D for preservation
- Affective appeal of the manatee
- QR Code = food safety

## Weaknesses

- Uncertainty regarding the consumption of cultivated meat;
- Neophobia;
- Structuring not yet well established;
- No clear definition of how royalties will revert to the endangered species;
- Will traditional local communities have access?
- Education and information about the product do not yet exist;
- Production steps are very specific, still difficult to scale;
- Illegal original meat;
- Much research still required for this particular species;
- No evidence of substitution for local consumption and how money will be reverted to conservation;
- Problem of preservation is not only financial;
- Price not yet competitive to the point of ending illegal hunting.

**Figure 2.** SWOT matrix presenting the main insights pointed out in the interviews.

Among the strengths of most interviewees, there is a social contribution in the sense that society can contribute to environmental development. In other words, conservation can be stimulated by the consumption of any citizen from anywhere in the world and, at the same time, by activating the curiosity of some consumers, they will become aware of what is happening to the cell donor. This could make consumers feel closer to the animal since it is not only a number, but a living being with a name and a history behind it. For instance, Baré and Kiniá are manatee individuals that were born in tanks, but later in life were released into the river. Scientists reported that the female was even found to be expecting a baby manatee. Here, we can highlight the fact that in a herd, no references to individuals exist, as animals such as cows and chickens only exist as a number. However, in this case, we have the aforementioned emotional appeal. All these aspects of the process can be shared through QR code information. It is not only a matter of eliminating the meat paradox, but creating a bond with an individual member of an endangered species, saving its life. Environmental conservation might be the major positive point of the model, as highlighted by the interviewees, which is the purpose of this business proposal: animal welfare.

Animal welfare is one of the main positive impacts of the project based on obtaining economic results from the exploitation of cultivated products. The manatee is part of a complex ecosystem within the Amazon rainforest, and preserving it would be fundamental for this reason alone. However, over the years, manatees have also become a symbol of successful models for preserving native Amazonian species. Based on positive economic results, it is possible to think of a causal nexus of improvement for the conservation of the species. In contrast with the problems caused by poaching, in the proposed model, the more cultivated manatee meat consumed, the better for the species.

Strengths linked to business opportunities go hand in hand with the growing demand for alternative proteins. At the same time that it has the power to encourage conscious consumption, this initiative provides an opportunity to expand this information, generating greater consumer awareness, as they have access to the QR code. This access can create an emotional appeal because buyers would see the animal alive and swimming. It should also be remembered that a certificate of origin is provided, which also guarantees food safety through the traceability access that the QR code provides. The model is economically viable, with a product that is theoretically easier to develop in terms of technology compared with shrimp and beef, reaching a niche market.

However, there are obstacles in the form of threats and weaknesses. It is worth noting that interviewees warned of the issue of the rebound effect, which would stimulate non-existent consumption of a threatened animal. This incentive would increase poaching. Even with reverted royalties from legalized sales of manatee CM, the Amazon region is too large for fully effective monitoring. Another issue raised by some of the interviewees was with regard to the local communities, bearing in mind that for them, eating wild meat is cultural. Furthermore, this new trade could become a black market activity, with differentiating between real meat and a cultured version of it constituting another issue. Previous studies regarding consumers' acceptance of novel food have shown that they present certain resistance to it [56]. As there is little information available about CM, people may not understand the entire process involved, which could create an obstacle to selling on a large scale.

## 4. Discussion

As mentioned above, a large and growing body of literature has investigated CM. However, no data have been found on the association between CM and wildlife conservation. Much of the research concerns consumers' perceptions of alternative protein, and there is a large volume of published studies describing the role of sustainability, environmental impact, and other concerns that connect CM to climate change mitigation policies. According to many in the field, it is only a matter of time before large amounts of alternative proteins are regularly purchased. However, because the present model is significantly disruptive, we first asked the experts if they viewed the proposal as feasible immediately after addressing the strengths, weaknesses, opportunities, and threats. At the end of each interview, we allowed all the interviewees to freely state their opinions regarding anything that they wished to bring to our attention. We also asked them what could be performed to improve the model. As a subcategory of analysis, with each one of the SWOT items, we identified a triple bottom line of sustainability, highlighting its social, environmental, and economic perspectives.

### 4.1. Feasibility of the Project

Technically, the engineers who were interviewed agreed that CM production is developing faster, but that there is still considerable room for improvement [15,57]. I1 stated that many different kinds of meat have been grown in bioreactors, including traditional livestock, such as chicken, pork, fish, and beef, but wild animals, to the best of her knowledge, have not. Therefore, much research remains to be conducted. I2 mentioned the structures of the tissues that would be easier in this model since there would be no "integration" between them, depending on the type of product desired (as the more minced the product

is, the less that integration is required). An example given by I2 was shrimp and salmon. These kinds of meat are usually consumed in whole cuts and have different structures to be built. Shrimps have an exoskeleton that is hard to copy, which is similar to salmon, with its fatness interspersed with lean flesh. Minced meat is an alternative to make the process easier. Since the same structure of meat does not need to be evolved, the texture may be different, but not the taste [58,59].

Regarding the acceptance of the project, marketing was cited by some of the intervie- wees as a strategic key to reaching consumers. I7 reported that "the main challenge with this kind of product lies in its introduction, which requires big marketing campaigns"; I4 stated that "We saw this in the plant-based market, even much more trivial, right? But they are effectively behind the marketing of large companies, they have chefs who simply serve to validate that something is edible, that it can be prepared, that it is a good catch [ . . . ]"; and I8 claimed that "It is very difficult for this public to be sensitized to consume this product. The strategy has to be directed towards those who have this focus—like ours, but with money. Let them pay for it. Create a media *fuzz*. You can buy people lunch in the middle of Rio de Janeiro for a tasting with all the publicity around it. Then you would have Bill Gates crazy about financing it".

Two interviewees with a background in research, I8 and I9, mentioned the importance of not treating consumers as ignorant, such as those who launched the technology of GMOs (genetically modified organisms), who argued that the population did not understand what was being conducted. I9 stated that "Transparency is the key". It is certain that the pros and cons of CM must be explained for a long time, as all disruptive innovations in the food sector have been in the recent past, such as margarine and sweeteners. Furthermore, one of the principles of the founders of alternative protein companies is transparency, precisely because they can defend proteins that are totally free from animal cruelty.

Regarding the reinvestments of the royalties, some respondents said that it was not clear to them how they would be converted to manatee conservation. This weakens one of the pillars of the model, which is precisely the positive connection between meat consump- tion and species conservation. More specifically, the interviewees from the conservation institutes agreed that the Amazon area was too vast to patrol. Therefore, the return of those royalties to monitor the entire animal habitat would not be manageable. These observations are supported by the previous literature regarding the rainforest [24,60].

### 4.2. The Opportunities and Strengths

Manatee CM can play a pivotal role in reframing the environmental impact of poach- ing in the Amazon region. Although manatee poaching might not be tackled directly, alternative products open up a new perspective of consumption, a new segment, a new market and, therefore, a new consumer. Niche consumers may embrace a new attitude regarding the manatee, separating it from illegal meat that is not safe and is consumed locally, viewing it as a product with high value and high levels of biotechnology. Opinion leaders consuming the CM of a wild animal that promotes sustainability, as in our proposal, might be seen as a positive action from the viewpoint of the public. This large mammal from the rainforest could have this appeal for consumers, according to I6. "They might be persuaded to buy something if they thought it was making a greater good", in the opinion of I5. In the words of I7, "preserving wildlife in the Amazon in particular is to the benefit of humanity". Some interviewees corroborated the assumption that this kind of publicity could strengthen the CM market.

Food safety could be another strength in this project, as cited by the interviewees, due to the QR code on the label and the production process. Certifying the meat's origin connects the product to the cell donor and operates in an inverse way to the usual cognitive dissonance used by the animal meat industry. For instance, people tend to experience a feeling of trust when a seal comes from a pro-environmental organization, as previously reported [61]. Substituting a real animal will sharply reduce the human–wildlife interface in the wildlife trade, which is a top priority for conservation, as demonstrated by Hilderink

and de Winter [61]. Disease transmission from wild and farm animals to humans is currently a major concern, with criminal networks and effects on local and global economies being pervasive [62]. Traceability can reduce these problems adequately. Food safety is also guaranteed by the type of industrial process, which is conducted inside bioreactors in an environment with controlled variables and reduced and manageable sources of contamination, in addition to the fact that there are no by-products in the process.

The experts who were interviewed also presumed that the unique experience and the storytelling that consumers would experience after eating an Amazonian delicacy would add considerable value to the product. According to I6, "in all ways this is a 'sexy' idea, with endangered species, cellular technology, and the Amazon".

Nevertheless, the main idea is the purpose of the business model, the aim of which is in keeping with most of the missions of alternative protein start-ups and is meaningful for humankind, the planet, and animals. Interviewee 8 mentioned that "almost all the CEOs of these start-ups, are mission oriented, people who were vegan or vegetarians, disgusted by animal cruelty, a whole series of things . . . you have a world of funding today, where you have... a world of investment funds, sovereign investment funds, you have an enormous amount of money that is directly intended for sustainability purposes".

*4.3. The Threats and Weaknesses*

Considering that the most important aim of the project is to enhance the protection of endangered species through the consumption of cell-grown products, a "highly unwanted rebound effect" (I6) is what concerned most of the interviewees, especially I3 and I11 from the conservation institutes, whose opinions prevailed against the idea as a whole. They claimed that the curiosity of some consumers for the real thing, instead of the cultivated version of manatee meat, might end up increasing poaching, leading to a decrease in the number of manatee samples. This would lead to the opposite of the expected effect, with greater consumption stimulating an increase in illegal hunting. "Current consumers of wildlife meat would only switch to eating cell-CM if: (a) the price was competitive; and/or (b) the risk and consequences of being caught and prosecuted for hunting or eating wildlife were sufficiently high as to be a deterrent" (I5).

I2, I8, and I10 drew our attention to genetic property rights and the registration of patents on manatee cells. Likewise, Fernandes et al. [63,64] highlighted that CM start-ups, at an advanced stage of production, are assigned patents, and multinationals in the food sector invest in CM start-ups, warning that over time, different stakeholders are emerging in this sector. Biopiracy is already a problem in the Brazilian Amazon, where genetic resources are traded with no regulation, and the resulting losses cannot even be quantified [65]. Because CM can be produced anywhere on Earth, any entrepreneur or start-up in the world can collect cells, basically from any animal, and proceed with the registration. If that happens, Brazilian producers of CM products would have to pay royalties on them, receiving no money to reinvest in the forest.

The current lack of a market for the product could be another obstacle, at least until such a market has developed (I5). Meanwhile, the lack of information available to consumers is also relevant (I7). In the last decade, the link between livestock and climate change has attracted a great deal of attention, but despite all the warnings, consumers remain reluctant to purchase less meat and continue to resist alternative proteins, as demonstrated in the literature [66]. This explains previous concerns regarding marketing with robust information.

I1 stated that CM is still under development; therefore, producing it in a scale amount involves the availability of many inputs, such as the basic scaffolds, chemical ingredients, bioreactors, and even professionals along the chain, which could be a problem. Few experts are prepared for this specific industry of CM, and even fewer for wild animal tissues.

The lack of legislation was another threat cited by I7 and I10. CM, up to the date of the data collection, was only available in Singapore, where the government had already regulated its consumption. The United States of America and the European Union were

discussing and analyzing this possibility in detail. On 16 November 2022, the FDA published an article saying that it had completed its first pre-market consultation for human food made using animal cell culture technology [67]. What is known about this kind of CM regulation comes from the accounts of observations of previous registration attempts. However, the bureaucracy involved in the registration of novel foods delays the process, and investors must be persistent until the goal of final approval is achieved.

As mentioned by I3, I9, and I11, manatee meat consumption dates back a long time in the Amazon, and previous studies have already linked hunting communities consuming bushmeat over domestic meat because of its cultural ties [68,69]. Therefore, these habits are difficult to remove from the local culture. Such actions might also affect local communities' historical traditions, threatening them in a social and cultural way, which might have serious consequences.

*4.4. Suggestions for Improvement*

According to the experts, manatee conservation helps to preserve a wide ecosystem of surrounding fauna and flora. However, surprisingly, manatee meat consumption has not yet been systematically researched, although it has been part of the culture of local communities for some time. There is very little evidence-based literature describing the impact of CM on wildlife consumption. Thus, work on this issue could result in new warnings for policymakers and conservation institutions and serve as suggestions for future and further research.

Wildlife conservation initiatives might also reduce the risk of new diseases, as interactions with humans will be reduced and the consumption of wild animals in nature will be replaced by food manufactured in industrial processes. A possible action is to provide local communities with an alternative source of protein [61]. I5 raised this possibility with the idea that for every can of manatee CM sold in the global market, one will be given to the local community in a system of "'buy-one, donate-one', in return for a no-hunting commitment from them". It is worth noting that this opinion gathered in the interview clashes with another piece of information collected, which is that the consumption of products in their most well-known presentation might encourage the capture of more individuals in nature.

Information, awareness, and consciousness were, at some point and in some form, mentioned in all the interviews. Most of the interviewees agreed that consumers must have previous education concerning the importance of conservation and conservation initiatives. In I7's words, "emphasis on marketing would be beneficial [ … ] Raising awareness through educational programs is important". This leads to another issue, which might be an opportunity for future research as well, the problem of approaching communities that still consume wild animals from Amazon fauna, aside from indigenous people. For the purpose of the present work, we did not look at the perception of consumers, as they were not interviewed. Likewise, hunters were also not interviewed. Gathering information from illegal activities involves risks that have to be managed, along with safety concerns for the interviewers. Likewise, Brazilian regulatory systems would not allow such interviews; thus, these groups were not selected to be among the interviewees.

The most important finding of this study is that consumer consciousness might not be prepared to accept the idea of the CM of wild animals, either because of the aforementioned rebound effect or the resistance to novel foods. With regard to the research question, the business model proposal could be a reality, and CM could help save the Amazon manatee and other species. However, this would require improvements, such as elaborating on the idea of royalties and mitigating the known rebound effects.

**5. Conclusions**

Our study provides a very disruptive narrative about the conservation of Amazon biodiversity through the consumption of the CM of wild animals from the rainforest. Combining efforts for wildlife protection and global concerns over food consumption, our model of-

fers insight for entrepreneurs wishing to invest in solutions, such as those mentioned above. The CM scenario continues to face many challenges but has promising opportunities.

A growing body of published work has provided new evidence on how to promote alternative protein production and consumption. These radical innovations in meat production and consumption are defining new strategies for sustainable protein purchase. Our contributions estimate the path that should be followed, opening new forms of pro-environmental businesses that spur sustainable development. Humans are facing life-changing opportunities that could bring an end to animal exploitation, and we could also reverse the biodiversity losses that have occurred in past decades. Due to the urgent situation that the planet is undergoing with climate change, poverty, and malnutrition, the Sustainable Development Goals can be achieved if efforts are made by different stakeholders.

Apart from the aspect of special care to avoid the rebound effect caused by any causal effect where the consumption of CM stimulates the consumption of fresh meat from the same animal, we were not able to capture in the interviews any major negative effects in the presented model. To mitigate this rebound effect, the dish chosen for the product can play a fundamental role. In the case of manatees, for example, developing the product in a presentation that is not well-known in traditional consumption, such as carpaccio or pâté, could have the desired mitigating effect.

The present study offers insights into this new phenomenon, which could potentially reduce the risk of food safety, food security, and the poaching of wildlife, while at the same time creating new jobs in a whole new market in the meat supply chain at the global level.

In conclusion, our work shows that cultivated meat has the potential to contribute to the transition to livestock-free meat production (with less GHG emissions). It brings attention to wildlife hunting and provides awareness to both cases, helping spread consciousness among consumers.

**Author Contributions:** Conceptualization: A.F.S.A., J.P.F.R., G.G.R. and A.C.; Methodology: A.F.S.A., J.P.F.R. and G.G.R.; Investigation: A.F.S.A., J.P.F.R. and A.C.; Writing—original draft: A.F.S.A., J.P.F.R., G.G.R. and A.C.; Writing—review & editing: A.F.S.A., J.P.F.R., G.G.R. and A.C.; Supervision: A.F.S.A., J.P.F.R., G.G.R. and A.C.; Project administration: A.F.S.A., J.P.F.R., G.G.R. and A.C. All authors have read and agreed to the published version of the manuscript.

**Funding:** This research received no external funding.

**Data Availability Statement:** The dataset presented in this article are not readily available because correspondence and requests for materials should be addressed to AFSA. Requests to access the datasets should be directed to AFSA: anaflavia.abrahao@gmail.com.

**Conflicts of Interest:** The authors declare no conflict of interest.

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
