# Peer review of "Cultivated Manatee Meat Aiding Amazon Biodiversity Conservation: Discussing a Proposed Model"

_conservation, doi:10.3390/conservation3020021_

Round 1

Reviewer 1 Report (Previous Reviewer 2)

Line 376 word spacing

Reference 1: Add DOI

References 28 and 47: revised DOI

Reference 57: delete the word "DOI"

Journals name in italics or not: 2. References must follow the same format, according to the journal's editorial standards.

Author Response

Thank you very much for your consideration. We tried to attend to all the suggestions, although some of the DOIs were not available for us to attach. 

Reviewer 2 Report (Previous Reviewer 3)

Please ensure that all literatures in the text have fully cited in reference list. 

Author Response

Thank you very much for your suggestion, we checked the references again and made the needed changes. We appreciate your time and consideration for this work of ours. 

Reviewer 3 Report (New Reviewer)

This study is, without a doubt, innovative and raises a series of pertinent questions in what will be new strategies to combat the loss of biodiversity. The authors state that this is a disruptive approach, something I absolutely agree with. It is, nevertheless, a relevant study and it's definitely worth considering for publication.
Although from a methodological point of view, few adjustments are necessary, I consider that this study has some flaws, namely in the apparent constant need to reinforce this idea as a future solution.

Introduction and literature review
It seems that these two sections repeat themselves in many arguments. Perhaps consider merging both and creating sub-sections such as the biodiversity crisis; the particular context of manatees; CM; CM alternative - benefits and challenges; etc. I believe there should also be an incursion into the cultural weight of manatee meat (as well as the species in question) among indigenous peoples, as this is not clear whether they are really a major factor in the decline of populations. Many examples of native wildlife exploitation exist that do not weigh heavily on the conservation scale.

Methods and data
It is not clear if they simply identified 11 experts or if there was a first group of more people and filtered down to 11. It is also not clear why the authors did not include other fundamental disciplines such as the social sciences for this discussion (e.g., human-wildlife conflict; sociology; etc.), since the topic involved the acceptance of new social concepts associated with conservation.
Although it assumes that this is the case, there is also no mention that the interviews were conducted separately.
It still lacks an explanation of why the manatee was chosen, as it is a species that has no global representation, a vulnerable status (if the authors are referring to Trichechus inunguis but, yet again, there is no clear information) and that leads to issues of unnecessary consumption (correctly mentioned by the interviewees).

Results
Much of the content of the results actually belongs to the discussion. There is a lot of interpretation to the results per se. Consider rephrasing this section to a more results-only approach.

Conclusion
The authors speak of animal suffering (line 543), a topic that has nothing to do with the subject of the study.

Author Response

Please see the attachmenT.

Reviewer 4 Report (New Reviewer)

The manuscript presents an evaluation of the idea of using manatee CM for the conservation of the species, and potentially the ecosystems this species lives in. The discussion proposed by the authors is relevant given the new wave of potential alternatives available to do conservation of wildlife and their habitats. The main idea of the paper is interesting, but my main concern is the limited number of interviews for the discussion and the lack of statistical analyses to examine the results. Eleven is a small number, and it would be highly desirable to obtain at least 20 interviews (it would have been really nice an analysis based on a contingency table). Still, even with 11 interviews, the authors could use simple quantitative analyses. Also, throughout the paper, the authors should also avoid expressions such as “much”, “many”, “most interviewees” or “Almost all the interviewees”, since such expressions may be misinterpreted; what means "much" or "many" for one person, may be interpreted quite differently by another. Providing a quantitative approach to the analyses and the presentation of the results will help the authors to deliver a more robust scientific contribution.

I suggest the authors to review, throughout the paper, the use of "preservation" and "conservation". Both terms are NOT synonyms and have different implications in the context of the discussion. As far as I can see in regards of the CM discussion, the possibilities for conservation of biodiversity are more likely than those for preservation of wilderness. I recommend the authors checking:  

-          Sarkar, S. 1999. Wilderness preservation and biodiversity conservation-keeping divergent goals distinct. BioScience 49:405-412.

In the Introduction is necessary to present the scientific name of the Amazon manatee. Using the vernacular name alone could lead to confusions. The authors should also consider adding some information on the ecology of this species that could help understand its importance in the Amazon ecosystems.

At the end of the Introduction the authors should present the particular research questions they are trying to answer. That will help the reader to understand the plan for the paper and will help the reader understanding the authors’ ideas.

The authors should seriously consider writing the whole paper in first person. Several papers have highlighted the advantages of doing so. An example on the matter

-          Lertzman, K. 1995. Notes on writing papers and theses. Bulletin of the Ecological Society of America 76:86-90.

Also, at the end of the Conclusions the authors should present a more solid take home message for the readers. Consider presenting not only a brief idea of what you did, but rather a solid statement of what was achieved out of the exercise.

As indicated earlier, I find the proposal of the paper interesting and worth discussing, and I hope my comments will help the authors to present a more solid contribution.

Specific comments

L. 5. Include the manatee’s scientific name.

L. 13-14. This sentence should be after the sentence in L. 8-10.

L. 50. Which burning are the authors referring to?

L. 199. Change to "These include experts in the fields of...."

L. 210-213. This information was presented earlier.

L. 222. Cite the “previous research”.

L. 228. Indicate whether this explanation was given by the same person.

L. 235-236. Is this an animal from a zoo that will never have the chance to live in the nature and it is used to blood samples and other similar procedures? Provide more details on how you visualize the process.

L. 243. Check the sentence.

L. 248.250. Check the sentence.

L. 263-264. Briefly explain each ones of the steps.

L. 277-280. Consider deleting this paragraph.

L. 285. Check the sentence.

L. 336-338. This point needs more explaining. The authors most make an effort to show why is it easier or cheaper to produce manatee CM than cow CM.

L. 344. Check the sentence.

L. 346-348. Move to Discussion.

L. 355-356. Cite at least the most important or relevant references.

L. 358-364. These are methods.

L. 370. Change “their” to “his” or “her”.

L. 376. Check spaces.

L. 404-405. This ideas needs further explaining.

L. 519-524. This information belongs to the methods section.

L. 528-529. Re-phrase. In the way is written it sounds like this strategy by itself could warranty the conservation of the manatee.

L. 542. Check sentence.

L. 545. Consider changing “climatic change” to "global changes".

Author Response

This manuscript is a resubmission of an earlier submission. The following is a list of the peer review reports and author responses from that submission.

Round 1

Reviewer 1 Report

The manuscript on cultured manatee meat aiding Amazon biodiversity preservation: discussing a proposed model is a very interesting paper. It also opens a new avenue of research in science for conservation and the economy. The manuscript is publishable overall, but the author needs to strengthen the English language.

Reviewer 2 Report

Comments

Delete space between words (Lines 88, 316, 356, 382, 383, 424, 443, 447, 461, 465, 482, 504, 522)

Improve the resolution of Figure 2

Line 170 Changed by Specth et al., 2018; Reis et al., 2020)

Line 72 Reis et al., 2020, 2021)

Line 180 Changed by Post, 2020, 2021)

Table 1 delete the hyphen of the Interviews

Line 243 Changed by (POst, 2012, 2014)

Add DOI in References 1, 14, 28, 47, 57

Reference number 14 changed "doi" by "DOI"

Reference number 30 add "DOI"

References 35, 46, 47, 51 The journal name without italics

Reviewer 3 Report

This study is interesting, and potential to be further researched. Unfortunately, authors only develop a study based on the opinion from experts (food and business), university researchers, policy maker and top level of businessman. It would be great if the study also gains more opinions from local people, or those who parts of the ecosystems where manatees are inhabited. 

Some threatens for the development of the CM of this species should be considered as parts of the problem to this product. It is also difficult to make a direct line between the CM products and preservation of the species as the weaknesses are highly influence the future plan of the program. However, there are also strength and opportunities that could be counted.

The paper raises urgent calls to all stakeholders for carefully pay attention to develop this CM product near a future.